# Metagenomic Insights into the Composition and Function of Microbes Associated with the Rootzone of *Datura inoxia*

**DOI:** 10.3390/biotech11010001

**Published:** 2022-01-14

**Authors:** Savanah Senn, Kelly Pangell, Adrianna L. Bowerman

**Affiliations:** 1Agriculture Sciences Department, Los Angeles Pierce College, 6201 Winnetka Avenue, PMB 553, Woodland Hills, CA 91304, USA; pangelkl4875@student.laccd.edu (K.P.); bowermal9924@student.laccd.edu (A.L.B.); 2Environmental Sciences Graduate Program, Oregon State University, Corvallis, OR 97331, USA

**Keywords:** weed science, plant-microbe interactions, medicinal plants, shotgun metagenomics, soil metabarcoding

## Abstract

The purpose of this paper is to elucidate the roles that microbes may be playing in the rootzone of the medicinal plant *Datura*
*inoxia*. We hypothesized that the microbes associated with the *Datura* rootzone would be significantly different than the similar surrounding fields in composition and function. We also hypothesized that rhizospheric and endophytic microbes would be associated with similar metabolic functions to the plant rootzone they inhabited. The methods employed were microbial barcoding, tests of essential oils against antibiotic resistant bacteria and other soil bacterial isolates, 16S Next Generation Sequencing (NGS) metabarcoding, and Whole Genome Shotgun (WGS) taxonomic and functional analyses. A few of the main bacterial genera of interest that were differentially abundant in the *Datura* root microbiome were *Flavobacterium* (*p* = 0.007), *Chitinophaga* (*p* = 0.0007), *Pedobacter* (*p* = 6 × 10^−5^), *Bradyhizobium* (*p* = 1 × 10^−8^), and *Paenibacillus* (*p* = 1.46 × 10^−6^). There was significant evidence that the microbes associated with the *Datura* rootzone had elevated function related to bacterial chalcone synthase (*p* = 1.49 × 10^−3^) and permease genes (*p* < 0.003). There was some evidence that microbial functions in the *Datura* rootzone provided precursors to important plant bioactive molecules or were beneficial to plant growth. This is important because these compounds are phyto-protective antioxidants and are precursors to many aromatic bioactive compounds that are relevant to human health. In the context of known interactions, and current results, plants and microbes influence the flavonoid biosynthetic pathways of one other, in terms of the regulation of the phenylpropanoid pathway. This is the first study to focus on the microbial ecology of the *Datura* rootzone. There are possible biopharmaceutical and agricultural applications of the natural interplay that was discovered during this study of the *Datura inoxia* rhizosphere.

## 1. Introduction

The *Datura* plant genus is of interest due to its production of several important cyclic compounds, most notably the tropane alkaloids. *Datura* produces alkaloids for anti-herbivory including scopolamine and atropine [1]. The plant contains antioxidants such as flavonoids, and phenols. Quercetin is a flavanol with anticancer properties that has free radical scavenging activity. Other compounds may include scopoletin [2], which is a fluorescent coumarin that has antifungal properties against antibiotic resistant strains of *Candida* [3].

This plant is sometimes referred to as an entheogen due to its use by native people in the United States such as the Chumash. *Datura* has been used for rituals and ceremonies in Hindu culture [4]. The seeds are highly heterozygous and spontaneous mutations are common in tissue culture [5].

A close relative of *Datura inoxia*, *D. wrightii* was recently recognized in rock art paintings by the native Chumash at Pinwheel Cave [6]. Although anthropologists long believed that the rock art paintings depicted a vision or hallucination, the recent discoveries paint a different picture. Rather than depicting a hallucination, the rock art depicts the sacrament itself and reflects the natural cycles of flowering and pollination that are essential to this dynamic outcrossing plant.

Since microbes from the *Datura* rootzone have been shown to suppress disease in other Solanaceae such as tomato [7], the distinct microbial community could be useful for biocontrol efforts. Recently, a few studies have elucidated the interaction between plants and microbes during secondary metabolite production. It is known that in some species, such as *Crotalaria*, nodulation is required for alkaloid production [8]. Furthermore, antioxidants have been shown to have antibacterial activity against pathogens [9], and antioxidants produced by soil microbes may have a plant-protective role. Functional analysis of soil metagenomic data has expanded as a field, and is used as a way to consider how plants are influencing the microbiome [10]. The microbiome is equally important in understanding plant secondary metabolism.

Shotgun metagenomics approaches in applied soil science provide the advantage of amplifying sequences from throughout the genome, including coding regions, yielding a high sequencing depth but at a higher cost. 16S metabarcoding is used as a diagnostic tool in clinical sciences and plant ecology, and its forté is identification of bacteria and archaea. However, metagenomics and metabarcoding are culture independent, meaning that no viable accessions are maintained. For that reason, microbial barcoding of streak plates isolated from the soil is also a useful tool that provides material for later wet lab experiments involving plant inoculation or antimicrobial trials, and helps to validate NGS results with tangible materials.

According to Sang et al. (2012), *Flavobacterium johnsoniae* produced 2,4-di-tert-butylphenol which had biocontrol activity against *Phytophthora* in *Capsicum* [11]. Sang’s study references one of a few studies in *Solanaceae* where *Flavobacterium* has been shown to suppress disease. Another example is Carrion et al. (2019), where *Rhizoctonia solani* was suppressed by the endophytic root microbiome [12]. Some of the most important natural products involved were phenazines, polyketides, siderophores, and chitinases. Aryl polyenes, which are structurally similar to carotenoids, were enriched by *Flavobacteriaceae*; terpenes and resorcinol were enriched by *Chitinophaga*. Similarly, in 2020 Lucke et al. reported that as biocontrol agents, the secretion of secondary metabolites triggered the induction of systemic resistance in plants [13]. In particular, they pointed out the importance of the chitinase from *Flavobacterium* and the Polyketide Synthase (PKS) gene cluster from *Chitinophaga* that is essential for disease suppression.

Del Barrio-Duque et al. (2019) found that 17/19 *Mycobacteriaceae* strains tested had fungal growth-promoting properties on *Serendipita indica* in vitro and in tomato; they were plant growth-promoting and helped to alleviate symptoms of *Rhizoctonia solani* [14]. In this research, *Mycolibacterium* was in a consortium with *Rhizobium* and *Paenibacillus*; in some instances, the isolations worked better than combinations of inoculants. Surprisingly, *Burkholderia* had a negative impact in vitro on *S. indica* fungal growth, although it has been shown to be plant growth promoting. An interesting review by Morris et al. 2019 highlights the fact that benzylisoquinoline alkaloid methyltransferases are functionally similar to the cyclopropane methyltransferase from *Mycobacterium tuberculosis* [15]. *Mycobacterium* cyclopropanation could contribute to tropane alkaloid biosynthesis in *Datura*, or provide an intermediate toward the same.

Banuelos-Vazquez reported that endophytes can receive nodulation genes, and in some cases, the ability to fix nitrogen, from *Rhizobium etli*. Plasmids are transferred to other endophytes in plant roots [16]. Another interesting example is from Kado and Kelly (2006), where they reported a successful protocol for transforming *Streptomyces* with *Agrobacterium* [17]. Lucke (2020) also noted the root nodulating capabilities and transfer of plasmids by *R. etli* and *A. tumefaciens*, and their ability to induce plant defenses [13].

Furthermore, the plant growth-promoting activities of endophytes may include production of plant growth regulators such as Indole Acetic Acid (IAA), and siderophore production for iron acquisition. *Streptomyces* spp. within *Bruguiera gymnorhiza* and *Boesenbergia rotunda* increased flavonoids and cyclopeptides with anti-HIV and anticancer activities [18]. Endophytic bacteria, as well as fungi, can produce paclitaxel [19]. Within *Taxus baccata*, *Streptomyces* produced 0.01–0.02 ng/mL, and *Bacillus subtilis* produced 1–25 ng/mL [19].

Indeed, as Wu et al. (2021) pointed out, the relationship between medicinal plants and endophytes is a long-term, symbiotic relationship [18]. Endophytes can strongly regulate the synthesis of secondary metabolites in plants. For example, for abiotic stress tolerance, in *Pteris vittata*, *Agrobacterium* and *Bacillus* spp. reduced arsenate to arsenite.

Volatile organic compounds can also help bacteria that are physically separated communicate with one another. Endophytic bacteria can protect plants by quenching quorum sensing molecules. For example, *Pseudomonas aeruginosa* could degrade 3-hydroxy palmitic acid methyl ester, which is a quorum sensing molecule of *Rhizoctonia solanacearum*. The inoculation of eggplant with *P. aeruginosa* reduced bacterial wilt caused by *R. solanacearum* [18]. In tomato, *Bacillus aureus* and *Serratia nematodiphilia* were applied to seeds along with *Ralstonia syzigii* sub-infection. The treatment under inoculation had increased jasmonic acid concentration in leaves and roots until 12 days post-infection [20].

Antioxidant enzymes are activated by endophytes, such as phenylalanine ammonia lyase, tyrosine ammonia lyase, and polyphenol oxidase (PPO) [20]. Rhizobacterial root colonization significantly impacted phenolic compounds, terpenes, and essential oils in plants. *Pseudomonas putida* root colonization altered benzoxaninone levels three days after inoculation. Colonization by *Rhizobium* changed the levels of phenolics, flavonoids, and anthocyanins, in blackberry, which was associated with delayed fungal postharvest growth [20].

In terms of the accumulation of plant secondary metabolites, production of plant sesquiterpenoids was enhanced in *Atractyloides macrocephala* by *Pseudomonas fluorescens*. Essential oil production was also increased [20]. According to Sang et al. (2012), endophytes can increase resistance to insects, while the plants provide nutrition and protection [11]. In *Nethapodytes fortida* and *Apodytes dimidiata*, two grasses, *Fusarium solanii* fungi were found to produce campothecin. This quinoline compound was previously taken from the roots of *Nethapodytes* directly.

The purpose of this paper is to elucidate the roles that microbes may play in the rootzone of the medicinal plant *Datura inoxia*. We hypothesized that the microbes associated with the *Datura* rootzone would be significantly different than the similar surrounding fields in composition and function. We also hypothesized that rhizospheric and endophytic microbes would be associated with similar metabolic functions to the plant rootzone they inhabited. This is the first study to focus on the ecology of the *Datura* rootzone microbiome, although similar studies have analyzed the rootzones of medicinal plants to uncover how microbes and their host plants interact during secondary metabolite production [21,22].

## 2. Materials and Methods

Soil samples were collected in triplicate from the field and arboretum at Los Angeles Pierce College. Replicated samples were taken from plant rootzones in the top 3 cm within a 1-foot radius of the plant subjects including *Alnus rhombifolia*, *Datura inoxia*, Ethiopian *Eragrostis tef* (Teff) grass, *Opuntia* cactus rootzone, and fallow conditions. The samples were collected in sterile cryotubes and were stored at −20 °C.DNA was extracted using the Qiagen (Hilden, Germany) Power Soil DNA kit and sent to Cold Spring Harbor Laboratory for 16S amplification, library preparation, and pooled 16S amplicon NGS on the Illumina (San Diego, CA, USA) MiSeq platform. The 515F (Parada)–806R (Apprill), forward-barcoded primers were used: FWD:GTGYCAGCMGCCGCGGTAA; REV:GGACTACNVGGGTWTCTAAT. A dual indexing strategy was used with these primers following Microbiome Helper’s approach [23,24]. Soil from the same sampling event was sent to Beijing Genomics Institute America, San Jose for logistics and processing, for the replicated *Datura*, *Alnus*, and fallow rhizospheric soil samples. These three plant rootzones were selected because *Alnus* and *Datura* are both known to produce antioxidants, and the fallow samples served as the reference. It was not possible to sequence the whole metagenome of all 5 categories due to budgetary constraints.

According to the USDA NRCS Web Soil Survey historic data, the fields related to the *Datura*, *Alnus*, and fallow samples are all classified identically as Cropley-Urban Land Complex 0–2% slopes, and the reported measurements for pH, texture, cation exchange capacity (CEC), and percent organic matter (% OM) were identical, as shown in Table 1 [25].

Whole genome shotgun sequencing, DNA extraction, and library preparation was subsequently performed by Beijing Genomics Institute on the DNBseq platform. Preliminary trim and QC were carried out using SOAPnuke [26]. MG-RAST (University of Chiago, U.S.) was used to generate functional and taxonomic profiles [27]. Taxonomic identification was performed using RefSeq (NCBI, Bethseda, Maryland, U.S.) and functional profiles were built from Subsystems identifiers. In order to determine which taxa were differentially abundant between categories, DESeq2 (EMBL, Heidelberg, Germany) was used in R [28]. STAMP (Dalhousie University, Nova Scotia, Canada) was used for functional analysis [29].

For soil metabarcoding and 16S barcoding analyses, DNA Subway was used which implements Qiime2 [30,31]. For soil metabarcoding, DNA extraction was performed with the Qiagen PowerSoil kit. Library preparation followed the Earth Microbiome Project 16S Illumina Amplicon protocol [32,33]. Sequencing was performed at Cold Spring Harbor Laboratories. The Greengenes identifier (utilizing 515F/806R primers) was used.

Bacterial isolations were also generated for the *Alnus*, *Datura*, and fallow rhizosphere samples in order to provide material for future in vivo plant experiments according to the Soil Science protocols published by St. Clair et al. [34]. Isolates from the soil solutions at 10^−3^, 10^−4^, and 10^−5^ dilutions were plated and grown for three days on various media, summarized in Table 2. Selected isolates were streaked on plates of the same media and colony PCR was performed after three days of isolated growth. For bacterial isolations, DNA was extracted by boiling colonies with chelex beads for 10 min, followed by 16S rRNA amplification. Amplicon sizes were between 500 bp to 1000 bp which allowed for putative identification of the isolates. Sanger sequencing for bacterial isolates was performed by Genewiz. QC was performed and consensus sequences were generated in the Cyverse DNA Subway Blue Line. The Phylip Maximum Likelihood tree was generated in Cyverse. Bacterial putative identifications for the isolates were generated using the EZBioCloud (CJ Bioscience, Seoul, Korea) 16S-based ID application [35]. The putative identification of the isolates with the short sequence length also were expected to reinforce the validity of the barcoding and WGS results, if the results were similar.

In order to test the response of bacterial isolates in broth culture of plant essential oils, two crude extracts were used. Lemon balm (*Melissa officianalis*) and Tea tree (*Maleleuca alternifolia*) essential oils are well-characterized; however, their interaction with rootzone microbes warranted further investigation. Vegetative plant material was harvested from and air dried at room temperature for one week. The samples were handled separately. Soxhlet extraction was carried out for three cycles using reagent ethanol as the solvent. The extracts were subsequently dewaxed by running them through a Buchner funnel and filter paper.

Bacteria were cultured in 1.5 mL tubes of nutrient broth for 48 hours at room temperature. Separate sterile glass tubes filled with 2mL of nutrient broth were inoculated with 100 μL of each bacterial broth culture. The glass tubes were inoculated with 20 μL of either lemon balm oil or tea tree oil from the soxhlet extractions. The clarity of the solution, which was a proxy for bacterial colonization levels, was measured as % transmission at 600 nm. Transmittance at 600nm was measured using a spectrophotometer after culturing at room temperature overnight. This assay was similar to Kryvstova et al.’s biolfilm assay with *Arnica montana* (2019) [36], although here cuvettes were used rather than microtiter plates. The data was analyzed using the R MASS package (University of Oxford, England) and visualized with ggpubr.

## 3. Results

### 3.1. Soil Samples WGS and MG-RAST Taxonomic and Functional Assignments

The sequence count ranged from 4,608,913 to 34,551,930 sequences. Interestingly the Fallow field had marginally higher observed alpha diversity overall (*p* < 0.1), which is contrary to what one might expect considering no plants were growing there. However, it was consistent with the metabarcoding results.

Figure 1 shows the top 14 bacterial families that were identified in a typical soil metagenome associated with the *Datura* rootzone and a fallow sample from this study. The remaining samples can be visualized in the published study on MG-RAST project mgp98747. All three samples exhibited similar proportions of *Mycobacteriaceae* and *Flavobacteriacea*; however, in the *Datura* rootzone *Sphingomonadaceae* sequences were more abundant. The *Cytophagaceae* sequences were represented in similar proportions in the *Datura* and *Alnus* associated samples, but were less abundant in the fallow samples.

In the *Datura* rootzone, some of the most abundant genera were *Pedobacter* and *Flavobacterium*. *Flavobacterium johnsoniae* is well-known for its tyrosine ammonia lyase gene, which has been cloned and used in genetic transformations to optimize production of aromatic compounds of pharmaceutical value in microbial expression systems. Some potential applications are production of tropane alkaloids and p-coumaric acid [37,38].

On the other hand, the *Datura* and *Alnus* samples had lower species diversity, suggesting that there may be some selection or recruitment occurring in the plant rootzones. The alpha diversity results are visualized in Figure 2. It is also interesting to note that the *Alnus* rootzone samples had less between group variation.

### 3.2. Functional Analysis in STAMP

Differential abundance of chalcone synthase genes was detected in the *Datura* rootzone based on STAMP analysis of the MG-RAST generated Subsystems metagenomic functional profile (Figure 3). Using a cutoff of alpha = 0.05, the following were significantly higher in the *Datura*-associated soil samples: Alpha-xylosidase (*p* < 0.017), chalcone synthase (*p* < 0.002) (Figure 3), putrescine utilization (*p* = 0.05), Aromatic amino acid transport protein AroP (*p* < 0.003), polyols ABC transporter permease component (*p* < 0.003), Quinate permease (*p* = 0.002), ubiquinone biosynthesis enzyme COQ 7 (*p* < 7 × 10^−3^), Heavy metal sensor histidine kinase (*p* < 6.4 × 10^−3^), flavin reductase family (*p* = 0.022), para-hydroxybenzoate- polyprenyl transferase (*p* < 0.004), Siderophore achromobactin ABC transporter (*p* < 0.0021), NADPH quinone oxidoreductase 2 (*p* < 0.0021), vanillate O-demthylase oxygenase subunit (*p* < 0.0014), Phenylacetic acid degradation protein PaaN (*p* < 0.0013), clavulanic acid biosynthesis (*p* = 0.027), cobalamin synthesis (*p* < 0.005), and MAP kinase pathways (*p* < 0.0045).

On the other hand, unlike the *Datura* samples, the Na+ translocating NADH-quinone reductase subunit was differentially abundant in Marquis C (*p* < 7 × 10^−4^), which was the fallow field near the compost pile. Gram positive cell wall components were differentially abundant in Marquis C (*p* < 0.008), although Gram negative cell wall components were equally likely to be present in all samples (*p* = 0.244). The CRISPR associated protein TM1812 was enriched in the fallow field near the compost pile (*p* < 0.0014). Phenlypropanoid compound degradation functional genes were most abundant in the *Alnus* samples (*p* < 6.4 × 10^−5^). Tocopherol biosynthesis genes were also differentially abundant in the *Alnus* rhizospere-associated soil (*p* < 0.004).

There was enrichment of cell wall degradation and depletion of Gram-positive bacteria, stress response, siderophores, catabolism of tyrosine, enrichment for aromatic compounds and permeases, production and reduction of flavonoids, antibiotics and cobalamin, and degradation of oxidized products in the *Datura* rootzone. The presence of differentially abundant permease components for essential oils and quinates suggests that there is a secretion pathway that would allow these compounds to enter plant roots and contribute to flavonoid, alkaloid and terpenoid biosynthesis in plant hosts.

### 3.3. Purple Line Analysis and Comparison

Demultiplexed sequence counts for the DNA Subway Purple Line paired end data set ranged from 5472 to 3.63 million, prior to alpha rarefaction. The lowest number of reads were for one of the *Datura* rootzone soil samples and a tilled fallow sample nearby in Marquis C. The highest number of reads were from Field 17, a fallow field tilled with cow patties and Marquis C near the compost pile. Indeed, in Field 17 also had a trend toward the highest alpha diversity, followed by Marquis C. There appeared to be more within group variation reflected by the 16S workflow compared to WGS.

Enrichment of *Pedobacter* matched up with MG-RAST results, and with *Mycobacterium* which may be related to alkaloid biosynthesis. *Pedobacter* represented >32% of the reads for one of the *Datura* rootzone samples. *Blastococcus* sequences were abundant in all three *Datura* rootzone soil samples. *Streptomyces*, *Rubrobacter* and *Mycobacterium* were abundant in one of the *Datura* samples, but were not detected in the other two. There appeared to be a similar drought tolerant entourage of bacteria associated with *Datura*, Teffgrass, and the cactus-associated soil samples, based on the Principal Coordinates Analysis (DNA Subway Purple Line Public project 7102).

Some unexpected results were detected by the isolation of bacterial colonies and analysis on the Blue Line. Four *Chryseobacterium* accessions were isolated from the soil that did not show up on 16S amplicon sequencing for the same soil samples. However, the *Chryseobacterium* genus was detected with DNBseq WGS (see Table 3). Since the WGS results are expected to be the most reliable, it seems likely that the *Chryseobacterium* putative identifications were in fact valid at least to the genus level, although that genus was not detected in the *Datura*-associated samples with metabarcoding in this instance. However, these results should not be interpreted to indicate novel species were isolates without further investigation, since the sequence lengths were too short to make a reliable identification at the species level. The Blue Line isolates were amplified using the same primers used for metabarcoding in this study and therefore have similar sequence lengths. In fact, the WGS results on the MG-RAST pipeline should be similarly considered reliable to the genus level.

### 3.4. Differential Abundance Analysis in DESeq2

Differential Abundance Analysis in R showed that there were a large number of taxa that were differentially expressed in the fallow (positive log fold change) versus *Datura* (negative log fold change) samples. There were 223 virus, archaea, or bacterial genera that were differentially abundant between the *Datura* rootzone samples and the Fallow samples from Marquis Field near the compost pile. There were 140 species that were associated with the *Datura* rootzone and 83 species that were associated with the fallow samples, based on adjusted *p*-value < 0.001. Of those, 124 genera were also significant based on the log fold change cutoff of 1.33, 34 were associated with the *Datura* rootzone, and the remaining genera were associated with the fallow samples The results are summarized in Table 3 and visualized in Figure 4.

Among the more notable results were differentially abundant bacteria in the *Datura* rootzone including *Flavobacterium*, *Chryseobacterium*, and unclassified members of the *Flavobacteriaceae*. *Chitinophaga*, *Mucilaginibacter*, and *Pedobacter* of the closely related *Sphingobacteriaceae* were also differentially associated with the *Datura* rootzone. These genera are associated with degradation of polysaccharides or chitin. There was also evidence that *Verrucomicrobium* that are typically associated with *Flavobacterium* as well as unclassified *Verrucomicrobium* were significantly associated with the *Datura* rootzone. These genera are known for hydrolysis of xylans. *Dyadobacter* of the *Cytophagaceae*, which has polysaccharide and amino acid degrading functions, was present in differentially large quantities; this species lives in glaciers and was formerly classified as a member *Flavobacteriales*. *Stigmatella*, which is responsible for breaking down insoluble debris, was also significantly associated with the *Datura* rootzone.

*Herbasperillum* and *Oxalobacter* from the *Oxalobacteriaceae* were differentially abundant, along with *Janthinobacterium*, which produces violacein, an antifungal, antiprotist, antibacterial, and anti-tumor compound [39]. Interestingly, sequences from the plant pathogen *Erwinia* were also present in significantly high numbers, as well as the poultry disease *Riemerella*, and the infectious *Serratia* which is consistent with the animal bedding that is spread from the compost pile to all of the adjacent fields. *Variovorax* is involved in disrupting quorum sensing and has swarming motility, and was differentially abundant in the *Datura* rootzone. *Bradyrhizobium* and *Azorhizobium* abundance in the *Datura* rootzone was significant based on adjusted *p*-value (*p* < 0.01) but not based on log fold change values.

### 3.5. Microbial Barcoding and Essential Oil Trial

The Phylip Maximum Likelihood phylogenetic tree indicated that there are four main clusters of bacteria that were isolated. The first cluster includes isolates related to *Pseudomonas* sp. The second cluster consists of two groupings; one group is related *to Massilia* sp. and *Achromobacter* sp., while the second group is related to *Chryseobacterium* sp. The third cluster consisted of *Bacillus* sp. and *Microbacteriaceae*. Interestingly, one of the isolates was putatively identified as *Priestia megaterium*, which is used as an herbicide in organic production [40]. The fourth cluster is made up of mostly pathogenic species such as *Erwinia*, *Pantoea*, and *Klebsiella*. Most of the pathogenic strains were associated with the Arboretum samples. *Stenotrophomonas* was used as the outgroup. There were four *Chryseobacterium* isolates that were detected with barcoding, which were not detected with 16S NGS metabarcoding. The ascertainment of the *Achromobacter* isolate agrees with NGS results. The *Xanthomonas* isolation agrees with the WGS results, since *Xanthomonadaceae* was the top family detected in the WGS samples for the *Datura inoxia* rootzone. There were several accessions with low similarity and high completeness which may warrant further investigation in order to more accurately identify the species. This could be achieved with a larger amplicon size or WGS of the isolates.

The main findings of the essential oil trials indicate that the isolate putatively identified as *Pseudomonas mucoides* was not susceptible to penicillin, and not susceptible to lemon balm essential oils. In retrospect, since *P. mucoides* is a Gram-negative bacterium, it would not be expected to be susceptible to penicillin. *Bacillus pseudomycoides* was resistant to penicillin but grew even more with lemon balm and tea tree essential oils. It is interesting because *B. pseudomycoides* was tolerant of high alcohol content, possibly since it is a lactic acid fermenter. *Priestia qingshengii* was not susceptible to penicillin and grew less when lemon balm or tea tree essential oils were added. *Leuthyella okanaganae* was resistant to streptomycin; however, growth was controlled significantly when lemon balm or tea tree essential oils were added and the least growth occurred when the tea tree treatment was used. Interestingly, there was an interaction between the essential oil used and the bacterial accession it was tested against, as shown in Figure 5.

It is interesting because Luna et al. 2007 found that several *Bacillus pseudomycoides* strains were susceptible to beta-lactams, however the strain ascertained in this study was resistant to penicillin [41]. *Bacillus pseudomycoides* is related to *Bacillus anthracis* and *Bacillus cereus*; these species are notorious for being antibiotic resistant but *B. pseudomycoides* and *B. cereus* are particularly resistant to clindamycin [41].

*Pseudomonas* is also notoriously resistant to antibiotics, although its inclusion here was somewhat serendipitous since it was not selected with an antibiotic that would be typically used to combat Gram negative bacteria. It was also found that tea tree and lemon balm oils were ineffective against the *P. mycoides* isolate.

## 4. Discussion

*Cytophagaceae* are known to degrade polysaccharides [42]. *Pedobacter* inoculation has been shown to significantly increase the antioxidant content of strawberries [43]. Closely related species are the *Flavobacterium*, which are famous for producing flavonoids including quinones and they play a role in disease suppression in soils [44]. Flavonoid and alkaloid production gene ontologies are closely related [45,46]. *Rhizoctonia solani* is suppressed by *Flavobacterium* [47]. This is believed to be related to its chitinase activity which is also of interest for energy production from biomass [48].

As Fadiji 2020 pointed out, plant protective endophytes stimulate plant secondary metabolite production while inducing plant resistance to pathogens [49]. In addition, endophytic fungi produce strong antioxidants. In *Taxus cuspidata* paclitaxel production was elevated, and in *Euphorbia pekinensis* there was elevated terpene production in response to the *Fusarium* E5 elicitor; these are two examples from the article. Some examples of protection provided by the endophytes include antibiotics, competition for resources with pathogens, and lytic enzymes such as chitinases that break down fungi and plant cell walls.

The chitinase activity provides a possible answer to how endophytes and their products are transported into the plant. Furthermore, xylanase and other cell wall degrading enzymes such as chitinases and cellobio-hydrolase were detected in the metagenome of several endophytes [20]. These lytic enzymes have the dual function of giving microbes access to plant cells, ducts, and intercellular spaces for colonization and transport, and the enzymes can also break down fungi cell walls to assist the plant host while fighting off infections.

Our preliminary evidence shows that the functional hits for Chalcone synthase in the microbes of the *Datura* rootzone are differentially abundant in contrast to the fallow sample. Chalcone synthase is involved in antioxidant production in bacteria and plants, and is homologous with polyketide synthase, which is part of the alkaloid production pathway in plants [50,51]. There appears to be synergy between the plant and microbe functions in terms of secondary metabolite production in the rootzone.

### Genetic Engineering and the Flavonoid Biosynthesis Pathway

Another way to look at this is that microbes may be engineered to produce the plant medicinal compounds directly [52]. Chemical synthesis of aromatic compounds uses benzene, toluene, and xylene as starting materials; these materials are derived from petroleum. Microorganisms are a renewable source of plant-derived secondary compounds. These compounds include phenolic acids, flavonoids, stilbenoids, coumarin, and conjugates of the same. A pathway of interest for the engineering of microbes for the production of plant metabolites, which occurs in nature, is the pathway by which tyrosine is converted to p-coumaric acid by tyrosine ammonia lyase and later converted to quercetin.

Yonekura et al. 2019 shed light on the flavonoid biosynthesis pathway [45]. In the plant kingdom, flavonoids are widely distributed in all subclasses, except for hornworts. This suggests that chalcone synthase, the first enzyme in the pathway, was evolved many times in parallel. If the enzyme was universal, then it must have been lost by many species along the way, since not all species within the subclasses have it [45]. The first two enzymes on the biosynthetic pathway are chalcone synthase and chalcone isomerase, which are believed to also have evolved first from a common ancestor involved in lipid metabolism.

There are 9000 flavonoid compounds defined strictly as:Structural derivatives of phenyl-substituted propylbenzenes with a C15 backbone,Phenyl-substituted propylbenzene derivatives with a C16 skeleton, orPhenyl-substituted propylbenzenes condensed with C6-C3 lignan precursors to form flavonolignans.

More broadly, we can also include chalcones and dihydrochalcones, anthocyanins, and aurones [45]. It is believed that defense against UV radiation, and plant hormone regulation were the first flavonoid functions. Flavonoid accumulation could help with harsh field conditions in *Datura*. They are structurally related to carotenoids, which are part of the accessory light harvesting complex in plants. Interestingly, UV-B radiation of mosses increased flavonoid content, which supports this notion [45].

Chalcone synthase (CHS) is derived from beta-ketoacyl ACP synthase. It is in the Type III polyketide synthase (PKS) gene family. CHS may have evolved before chalcone isomerase (CHI), since CHS catalyzes the first step and the step catalyzed by CHI may happen spontaneously [45]. P-coumaroyl CoA is transformed to naringenin chalcone by CHS, then to naringenin by CHI. Arabidopsis has four Type III PKS genes and one functional CHS.

CHS is promiscuous in the sense that it can use many substrates, although it cannot function on bulky substrates [45]. *Flavobacterium johnsoniae* has a chalcone synthase gene according to the Uniprot database [53], which may be working synergistically with *Datura* to increase flavonoid content in the plant, supply acyl compounds to the plant for downstream production for plant secondary metabolites, or have a modified function that is specific to its plant host. It is worth mentioning that *Datura*, *Streptomyces*, and *Agrobacterium* also have this gene. In vitro, CHS from *Huperzia serrata* can produce other polyketides besides flavonoids, like aromatic tricyclic pyridoisoindoles [45].

Evolutionarily, CHS on the flavonoid biosynthetic pathway may have paved the way for biosynthesis of medicinal secondary metabolites by land plants. It expanded the chemical variety of the metabolites plants produced over time [45]. Type III PKS can also synthesize oxidosqualane and terpenes in other specialized metabolisms, for example. It is not known why plants and their endophytes have evolved this important set of enzymes, but it warrants further investigation.

Many medicinal plants produce economically important compounds, and synthetic production is not possible because the pathways are poorly understood. Further elucidating the interaction between microbes and wild plants could also shed light on resilience mechanisms, which could be used for crop improvement. For example, it has been shown that beneficial microbes prime plant defenses, giving a competitive advantage to organic farming [54]. Furthermore, understanding how these plants adapt to their environment will empower weed control efforts, including biocontrol.

## 5. Conclusions

We hypothesized that the microbes associated with the *Datura* rootzone would be significantly different than the similar surrounding fields in composition and function. We found strong evidence to support this hypothesis, based on significant Log fold change values and adjusted *p*-values. The main results indicated that *Flavobacterium, Pedobacter* and *Chitinophaga* were differentially abundant in the *Datura* rootzone when compared with the fallow condition. We also hypothesized that rhizospheric and endophytic microbes would be associated with similar metabolic functions to the plant rootzone they inhabited. There was some evidence to support this notion, based on experimental evidence and statistical analysis of the functional profiles. One of the soil bacterial isolates, putatively identified as *Bacillus pseudomycoides,* grew more when treated with lemon balm and tea tree essential oils. There were several functions associated with flavonoid and aromatic compound production, as well as cell wall degradation, that were differentially abundant in the *Datura* rootzone, compared to the fallow condition. Further research focusing on metabolomic profiles of *Datura* plants grown in the presence and absence of rootzone bacteria will be useful to reveal to what degree rootzone bacteria influence secondary metabolite production.

## Figures and Tables

**Figure 1 biotech-11-00001-f001:**
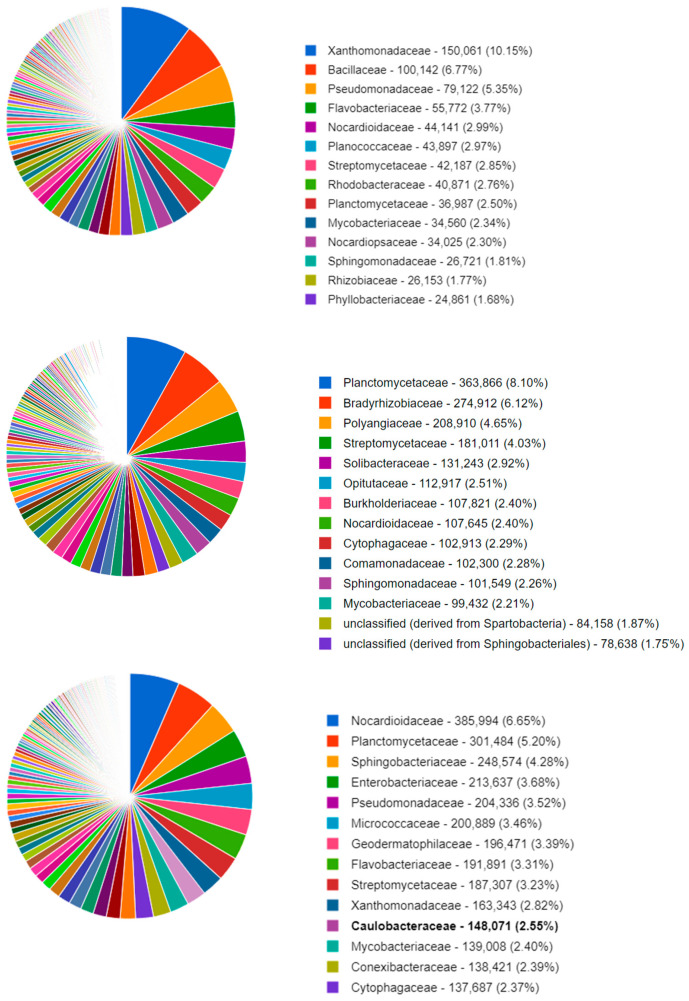
MG-RAST classification of the NCBI RefSeq identification of the soil organisms at the family level, for a representative sample of fallow (**top**) *Alnus* (**middle**), and *Datura* (**bottom**) rootzone soil.

**Figure 2 biotech-11-00001-f002:**
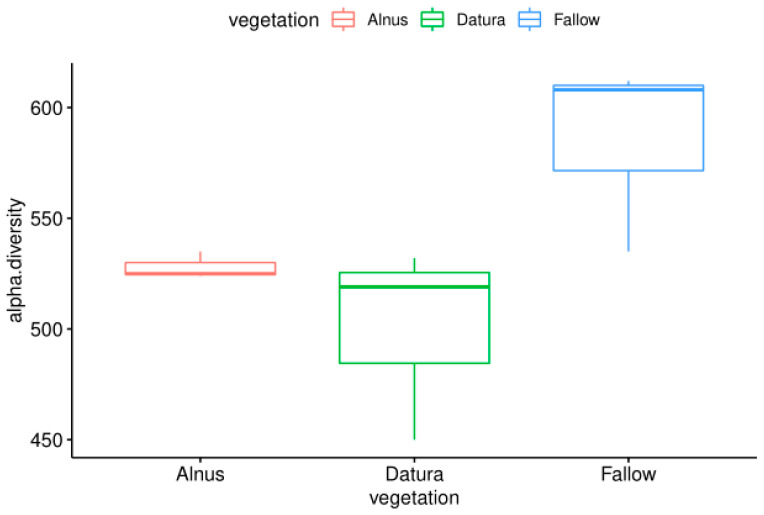
Comparison of the observed alpha diversity levels from fallow, *Alnus*, and *Datura* rootzone-associated soil samples from the Pierce Farm in Los Angeles, California.

**Figure 3 biotech-11-00001-f003:**
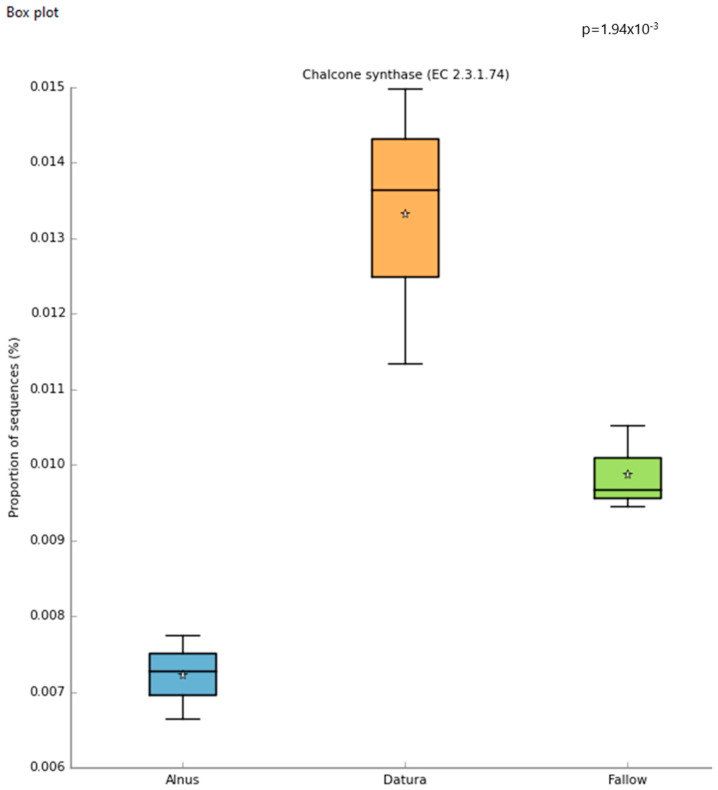
Boxplots of the MG-RAST generated Subsystems metagenomic functional profile give insights into the community functions that are most significant. The star symbol represents the mean proportion of sequences for each sample group.

**Figure 4 biotech-11-00001-f004:**
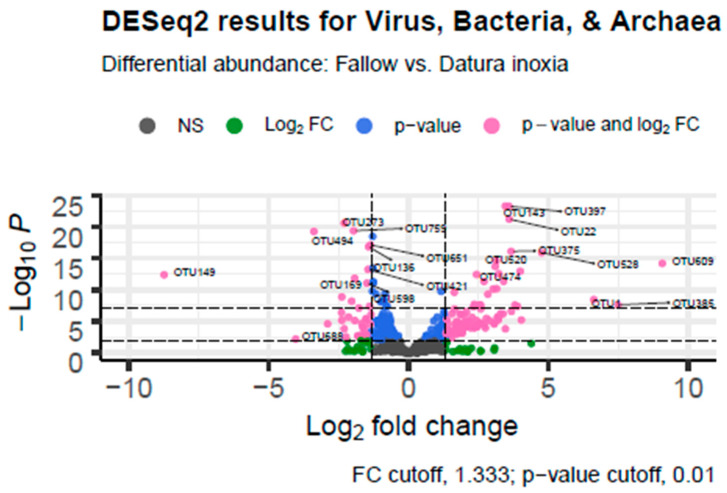
Differential Abundance Analysis Volcano Graph. DESeq.

**Figure 5 biotech-11-00001-f005:**
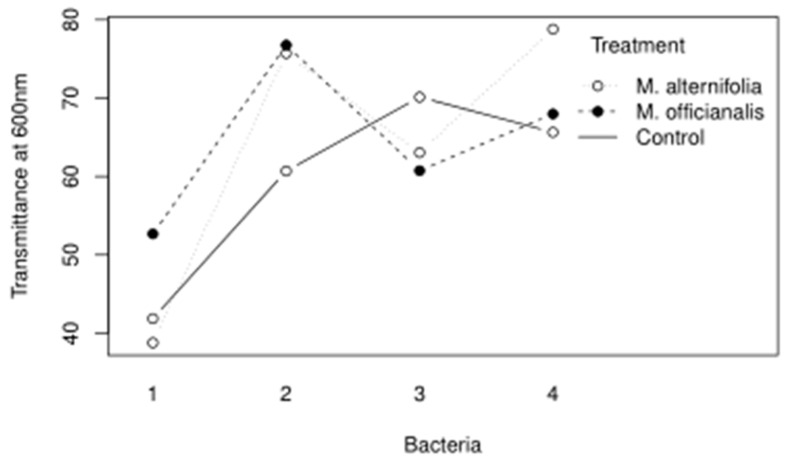
The interaction plot for the Bacteria and Treatment variables of the essential oil trial are shown.

**Table 1 biotech-11-00001-t001:** Selected physiochemical properties of soil.

Map Unit Name	pH	CEC	% Sand	% Silt	% Clay	% OM
Cropley-Urban Land Complex, 0 to 2%	7.9	37.5	22.1	27.9	50	1.5

**Table 2 biotech-11-00001-t002:** Bacterial Strains Isolated and Culture Conditions.

Sample No.	Field	Colony Color	Morphology	Antibiotic Used	Media	Absorbed CR
1	Marquis C	Transl yellow	Unkown	NA	Nutrient agar	NA
2	Marquis D	White/tan	Unkown	Penicillin	Nutrient agar	NA
3	Field 28	White/tan	Unkown	Penicillin	Nutrient agar	NA
4	Marquis A	Yellow	Unkown	Streptomycin	Nutrient agar	NA
5	Marquis A	Yellow	Raised	NA	Nutrient agar	NA
6	Arboretum	White	Spreading	NA	TYES-CR	absorbed cr
7	Marquis A	Transl yellow	Raised	NA	Nutrient agar	NA
8	Marquis A	White	Gliding	NA	Nutrient agar	NA
9	Marquis A	Transl yellow	Gliding	NA	Nutrient agar	NA
10	Marquis A	Yellow	Flat	NA	Nutrient agar	NA
11	Marquis A	Transl Yellow	Highly motile at 4 °C	NA	Nutrient agar	NA
12	Arboretum	Clear/tan	Mucoid	NA	TYES	Intense pink
13	Arboretum	White	Flat	NA	TYES	Dark red
14	Marquis A	Yellow	Raised	NA	Nutrient agar	NA
15	Arboretum	Clear/tan	Small mucoid	NA	TYES	Light pink
16	Marquis A	Clear/tan	Mucoid	NA	TYES	NA
17	Marquis A	Transl yellow	Mucoid	NA	TYES	NA
18	Arboretum	Clear/tan	Gliding	NA	TYES	Abs cr
19	Arboretum	White	Spreading	NA	TYES	Abs cr ring
20	Marquis A	Transl yellow	Raised, wrinkled	NA	1/2 NA + AC	NA
21	Marquis A	Transl yellow	Small colony	NA	1/2 NA +AC	NA
22	Marquis A	Transl yellow	Small colony	Griseofulvin	ISP-6	NA
23	Marquis A	Yellow/orange	raised	Griseofulvin	ISP-6	NA
24	Marquis A	White/grey	Large colony	Griseofulvin	ISP-6	NA
25	Marquis A	Transl yellow	Wrinkled	Griseofulvin	ISP-6	NA

**Table 3 biotech-11-00001-t003:** The 34 genera with both high negative log fold change values, and significant adjusted *p*-values (*p* < 0.001) representing those associated with the *Datura* rootzone are shown.

OTUID	Genus	log2FoldChange	*p*-adj.
OTU149	Cavemovirus	−8.74069944	1.46 × 10^−^^11^
OTU494	Pantoea	−3.382739635	5.9 × 10^−18^
OTU168	Chryseobacterium	−2.894493584	0.000172
OTU332	Janthinobacterium	−2.396801275	5.53 × 10^−6^
OTU307	Herbaspirillum	−2.385130454	2.86 × 10^−8^
OTU505	Pedobacter	−2.384332893	6 × 10^−5^
OTU247	Erwinia	−2.324319999	3.84 × 10^−35^
OTU738	unclassified (derived from Flavobacteriaceae)	−2.299754251	0.00078
OTU273	Gemmata	−2.299386567	3.76 × 10^−19^
OTU308	Herminiimonas	−2.1461305	2.3 × 10^−5^
OTU693	Variovorax	−2.033564378	1.25 × 10^−7^
OTU759	unclassified (derived from Verrucomicrobia subdivision 3)	−1.966909323	5.46 × 10^−18^
OTU169	Chthoniobacter	−1.924054848	5.19 × 10^−11^
OTU440	Mucilaginibacter	−1.920113231	6.47 × 10^−5^
OTU263	Fluoribacter	−1.819462381	0.006919
OTU569	Riemerella	−1.783874932	0.008796
OTU488	Oxalobacter	−1.775720869	1.2 × 10^−6^
OTU339	Klebsiella	−1.680008901	0.00538
OTU277	Geodermatophilus	−1.67277933	0.000157
OTU463	Nitrospira	−1.638446217	0.000467
OTU262	Flavobacterium	−1.633855026	0.007125
OTU156	Chitinophaga	−1.588995265	0.000676
OTU696	Verrucomicrobium	−1.523905486	3.61 × 10^−5^
OTU234	Dyadobacter	−1.505907596	0.005228
OTU480	Opitutus	−1.500266429	5.62 × 10^−5^
OTU598	Serratia	−1.489698167	2.49 × 10^−10^
OTU421	Methylobacterium	−1.448043272	2.59 × 10^-12^
OTU136	Candidatus Solibacter	−1.428079685	1.1 × 10^−15^
OTU624	Stigmatella	−1.392596868	6.52 × 10^−7^
OTU751	unclassified (derived from Proteobacteria)	−1.373840383	0.003172
OTU180	Conexibacter	−1.373047754	0.000188
OTU651	Terriglobus	−1.362530369	6.27 × 10^−16^
OTU532	Polaromonas	−1.340071098	9.35 × 10^−6^
OTU727	unclassified (derived from Candidatus Poribacteria)	−1.330571533	0.00172

## Data Availability

WGS profiles, metadata and genomic data are freely available on MG-RAST under public project mgp98747. DNA Subway Purple Line metabarcoding results are available on Cyverse under public project 7102. Raw fastq files are available at: https://de.cyverse.org/data/ds/iplant/home/shared/iplant_DNA_subway/metabarcoding_data/210611_M00970/St_Claire (accessed on 24 June 2021). DNA Subway Blue line barcoding results and trace files are available under public project 238876 on Cyverse.

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
