# Peer review of "Metagenomic Insights into the Composition and Function of Microbes Associated with the Rootzone of Datura inoxia"

_biotech, 2022, doi:10.3390/biotech11010001_

Round 1

Reviewer 1 Report

The research is very interesting and concerns multi-faceted and difficult laboratory work. The topic fits well with the magazine's interests and is fresh. The role of root zone microorganisms is new and inquisitive. The authors used microbial barcodes to test essential oils for antibiotic-resistant bacteria and other soil bacterial isolates, next-generation 16S sequencing (NGS), and genome-wide taxonomic and functional shotgun (WGS).
The title of the work: Microorganisms in the root zone of Datura contribute to supporting the antioxidant system of flavonoids and other aromatic compounds, which is not confirmed by the results of the research. The figures and tables show the soil microbial biomass. The authors hypothesized that rhizospheric and endophytic microorganisms capable of performing the same secondary metabolic functions in the root zone of the plant they inhabited would be found. They hypothesized that microbial functions would influence and support the production of plant secondary metabolites, for example by providing precursors to important plant bioactive molecules. However, the results of this work do not support this hypothesis. The considerations of the authors are based on extensive references, and not on the results of their hard laboratory work.
Therefore, I propose to change the title and purpose of this article solely on the basis of the results of the work.

The article is well written and has good references, but has a few bugs:
without parentheses date of publication of the work next to the name of the first author
lines 121-122 Kado and Kelly (2006)
Poems 123-124 Lutsk (2020)
lines 132-133 Wu et al.
lines 155-156 According to Sang et al. (2019)
line 321 Fadiji (2020)

Names of Tables and Figures require correction not only of the font, including:
Table 1. Selected physicochemical properties of soil (most often in the tables, the authors present the content of TOC, NT)

Figure 6. Shows the interaction graph for the Bacteria and Treatment variables in a sample of essential oils. Interestingly, there was an interaction between the essential oil used and the appearance of the bacteria against which it was tested.
The second sentence is not an image title but a commentary and should be included in the discussion section. The drawing is not clear and gives the name of the treatment.

Line 219 Data was analyzed using R language, which is not clear enough, please provide more information.

Author Response

Thank you for your comments.

I have considered your comments in regards to the title and the purpose, and I have updated the title, abstract, introduction and conclusion to be more focused on the statistical tests performed and the results of the study.

The problems with in-text citations have been corrected.

Table 1 Name of the table has been updated. Please note that %OM is the same as the total organic carbon percentage. 

Figure 6 The title of the figure has been updated. The figure legend was updated to be more clear about the treatments.

For the data analysis portion of the material and methods section, more information was provided in this revision. 

Reviewer 2 Report

This manuscript suffers with serious flaws. It look like undergraduate student project with out scrutiny has been submitted as manuscript. 

Introduction section too long it has to be reduced. Avoid subheading. Lot of information on plants, it should be avoided.

In Methodology: Sampling and sequencing was not clear.

cactus and teff grass samples are missing in Beijeing Genomics Institute

What is the purpose of going two metagenome one at Cold spring antoher at Beijeing Genomic?
where is alpha and beta diversity?
You should display rarefaction curve.

Fig 4 is not essential and does not convey any meaning
You should also perform diversity indices.

Cultivable bacteria:

What is the media used for isolation? On what basis these bacteria were selected?
One bacterium from one sample? it is hard to believe. Low completeness reflects on reduced similarity similarity percent.

L 203: must be more than 1500bp; 500 bp is too low length very hard to identify the bacterium.

I advice authors to resequence the all the isolated bacteria.

Table 1 is not necessary. it can be given in running text. 

In Result: it is poorly explained. 

Discussion section: Why you start with CHS and focussing on CHS alone. It does not make any sensse

Author Response

Thank you for your comments.  

The introduction section has been edited, to be shorter and contain more focus on bacteria and less information about plants. Subheadings were deleted.

More details about sampling and sequencing methodology have been added.

Please see lines 102-120 which explain why the selection of methodologies was used, i.e. one at Cold Spring and one at BGIA.

The alpha diversity graph is now included in the article as Figure 4.

The original figure 4 has been deleted.

The alpha rarefaction is discussed beginning with line 316, under Purple Line Analysis.

Please see the list of media used for cultivation in table 2.

Please see new comments included in the article starting on line 395 about the issues you mentioned with the 500 bp amplicon size.

The results section has been edited in order to make the results more clear.

In the discussion section, chalcone synthase was focused on because it was one of the main genes that was differentially abundant in the Datura rootzone. After editing the abstract and adding a conclusion, I hope that this context makes more sense in the discussion. 

Reviewer 3 Report

Review Comments:
Title: Microbes in the Datura rootzone contribute to an antioxidant support system of flavonoids and other aromatic compounds
Remarks:
The manuscript entitled “Microbes in the Datura rootzone contribute to an antioxidant support system of flavonoids and other aromatic compounds” is written well. The review is ready to be published in its current form but can be considered after the minor amendments. The content of the research is enough and supported by enough reference evidence. I would recommend this paper be accepted after major revision. Following are my specific comments to further polish the manuscript:
General Comments -
• Title – Acceptable.
• Abstract – Abstract is not properly organized. Need to be improved. Include the obtained results. Novelty line?
Line@17 check the italics? “and” should not be italic. Also check the species names. If only genus mentioned, it should be normal. If you include the species, then only need to be italic. Example: Cyanobacteria/ Cyanobacteria sp.
• Keywords – Acceptable.
• Introduction – The first paragraph nor referred with any citations?
Add some introduction on current and future status. Please refer to similar articles to revise them properly. The novelty of this work? Explain?
Line@34 Some typos
Line@36 Datura has analgesic properties. This line too general.
Authors made some subtopics in introduction. According to my suggestions all the introductions can be kept under 1 Topic that is introduction.
Line@146-152 Following references may help to improve this section
- Antioxidant and antibacterial activity of red seaweed Kappaphycus alvarezii against pathogenic bacteria. Global Journal of Environmental Science and Management, 6(1), 47-58
• Material and methods: This part can be written in sub section.
- Sample collection
- DNA extraction
- NGS
- Analysis
• Results and discussion: Authors discussed the results impressively. This part is
acceptable. The sub sections mentioned are good enough.
Line@276 Marquis C. means? Italics? Please check the species names and correct them.
Figure@4 Not clear unable to read the tree branches. Provide high quality figures.
• Conclusion – I am unable find this section? Please provide the conclusion, if necessary,
by journal guidelines. Same goes to figure 5 and 6.
Minor Specific Comments –
1. Follow the author guidelines properly.
2. Superscripts need to be checked.
3. Include high-quality graphs and figures.
4. More recent references need to be included from 2020 to 2021.
5. Some grammatical mistakes should be corrected throughout the manuscript.
In the end, I would like to summarize those above corrections that should be done to improve the
overall quality of the manuscript.

Author Response

Thank you for your comments.  

The abstract has been improved to be more organized and focused on the main results of the paper. There is now a novelty line.

The introduction has been improved to include additional references about the current and future state of the topic, and the novelty of the work is mentioned.  Some typos were deleted and some information that was too general was deleted from the introduction. Subtopics were deleted. 

Thank you for your comment about the red seaweed article. This reference is now included in the paper. 

Species names were corrected and low quality figures were deleted or replaced.  Some grammatical errors were resolved.

A proper conclusion has been provided in the new revised version. 

More references were added, including recent references from 2020 and 2021.   

Reviewer 4 Report

Conclusion is missing in the manuscript. It should be included at the end of the study clearly depicting the significance & outcome of the study undertaken by the researchers.

Author Response

Thank you for your comments.

The presentation of the results has been improved by revising the text for clarity, and replacing or deleting low quality figures.

A proper conclusion has been provided in the new revised version. 

Round 2

Reviewer 1 Report

The authors presented an updated version of the software
But I see mistakes:
line 113; should be Wu and v. (2001)
table 1,2,3 the same rule
The title of Figures 3 and 4 to the sentence from the paragraph or methodology is not a title

Author Response

Thank you for your comments. 

The titles for figure 3 and Figure 4 was updated using captions from similar articles as a model.

Table 3 has been eliminated from the text. Table 1 title was updated according to another reviewer's recommendation.  Table 2 title was not updated because it was the most concise way to present the information. 

The citation at line 113 was updated according to your recommendation. 

Reviewer 2 Report

Though the revised version improved but most the technical flaws are pointed out in my previous review is not addressed adequately. For detailed comments please refer attached pdf file. I have made corrections in stick note.

Author Response

Thank you for your comments, which have improved this paper.

Note the following changes were made or the explanation why they were note made:

In several instances the abbreviation for species was italicized. That has been corrected according to your recommendation.

Several grammatical errors or unclear phrases were corrected according to your recommendation.

There was concern about the identifications given to the species level in Table 3. Therefore additional discussion was added to the paper about the limitations of the 16S IDs and the degree of their validity (lines 358-368), and all identifications were referred to as putative throughout the paper.

The table with the differential abundance analysis, formerly Table 4, and now relabeled as Table 3, is intact. I prefer to keep this table in the article to make it easier for future researchers referencing the article to see the main results, other than reading a narrative or list. 

For the same reason, I prefer that Table 1 remain in the text to give a fast reference for the soil characteristics. Also, other reviewers thought it was important.

Figure 1 was improved by including the main results for all three of the soil profiles at the family level, according to your recommendation.  

Reviewer 3 Report

The authors improved the revised manuscript according to raised comments. I recommend present manuscript can be accepted for publication.

Author Response

Thank you for your comments, which have improved this manuscript.